# OpenReview forum: "AgentTuning: Enabling Generalized Agent Abilities for LLMs"
_ICLR.cc/2024/Conference — Submitted to ICLR 2024_

### Official Review · Reviewer_iTiT · 2023-11-01

**Soundness:** 3 good
**Presentation:** 3 good
**Contribution:** 2 fair
**Rating:** 6
**Confidence:** 4

**Summary:**

This paper studies mixing state and action trajectories with general instruction tuning corpus to improve open-source LLM agents’ decision making capabilities while keeping their generalization performance. The authors propose a 3 stage approach: generating instructions, collecting trajectories, and filtering them based on task reward. For tasks where instructions are not already given, the paper uses similar auxiliary tasks or zero-shot prompts the LLM to construct a task input and outputs. Trajectories are collected by prompting the LLM agent using ReAct to generate actions, running in the environment to collect rewards and extra feedback, and continuing in a loop till an episode is terminated. Trajectories are filtered based on task specific thresholds. Finally, this trajectory corpus is combined with ShareGPT corpus to train a Llama-2 model. The authors show that the model improves baseline Llama-2 on decision making tasks while keeping its general capabilities and is competitive with GPT-4.

**Strengths:**

The paper is written well and easy to follow. It presents a set of expensive experiments, showcasing that open-source LLMs can be competitive with proprietary LLMs when trained on the right data.

**Weaknesses:**

While the empirical contribution is significant, the paper overall feels incremental with straightforward improvements over prior instruction tuning and knowledge distillation. Some of the design decisions are also not explained.

1. While the agent trajectories are very valuable and costly to collect, they are mainly extracted from public tasks/benchmarks by using ReAct with GPT models. The overall process with instruction generation, trajectory collection, and filtering can be useful for other data collection efforts but they are relatively straightforward. For example, ReAct/COT prompting is used with no significant change, reward filtering is also a standard practice in imitation learning. I suggest highlighting main challenges and how significant they are in addition to remedies that you introduced.

2. Using an already available corpora with ReAct and filtering trajectories based on a threshold seems to be reducing the size of the data drastically. For example, in Mind2Web, only a tiny fraction is kept. It is not clear if the benefit of COT annotated trajectories can overcome the reduction in the data size. Can you present results where you make use of the data as much as possible even if you can’t inject COT annotations?

3. How did you choose $\eta$? Given that it denotes the tradeoff between broader generalization vs agent-task performance, it is important to highlight its impact. Are your models sensitive to this parameter?

4. Similarly, how did you decide 1:4 ratio for GPT-4 vs GPT-3.5?

5. What is GPT-4’s interaction trajectory with the DB? How did you end up collecting multi-turn dialogues? Are you using rule-based turn generation to convert DB response into a prompt?

6. Is ShareGPT not leaking any of the test-time task data? It would be helpful to clarify.

**Questions:**

1. What are main challenges that you had and how significant they are?

2. Can you make use of the available data as much as possible? Would that improve the results even without the COT annotations?

3. How did you choose $\eta$ And 1:4 ratio?

---

> ### Author Response · Authors · 2023-11-20
> **Response to Reviewer iTiT (1/4)**
>
> Thank you for your insightful comments and constructive feedback on our paper. Your questions have provided us with an opportunity to clarify certain aspects of our work, which we hope will enhance the overall quality and understanding of our research.
>
> > Weakness 1: While the agent trajectories are very valuable and costly to collect, they are mainly extracted from public tasks/benchmarks by using ReAct with GPT models. The overall process with instruction generation, trajectory collection, and filtering can be useful for other data collection efforts but they are relatively straightforward. For example, ReAct/COT prompting is used with no significant change, reward filtering is also a standard practice in imitation learning. I suggest highlighting main challenges and how significant they are in addition to remedies that you introduced.
>
> We appreciate your insights regarding the perceived straightforward nature of our methodology. However, we would like to emphasize some unique aspects and significant contributions of our work:
>
> - **Limitations of prior works:** Existing studies on LLMs as agents have thus far largely focused on designing prompts or a framework for completing one particular agent task [1][2], rather than fundamentally enhancing the agent capabilities of the LLMs themselves. In addition, many efforts are dedicated to improving LLMs in specific aspects, involving fine-tuning the LLMs using datasets tailored to specific tasks [1][3]. This overemphasis on specialized capabilities comes at the expense of the LLMs’ general abilities and also compromises their generalizability.
> - **First General Agent Model to Match GPT-3.5-turbo:** As illustrated in Figure 1(b), there is a substantial gap between open-source and API-based models like GPT-3.5 and GPT-4. Our work introduces the first open-source model, AgentLlama-70B, which **matches the capabilities of GPT-3.5-turbo in general (unseen) agent tasks while preserving its general abilites**. We believe this makes a significant contribution **both academically and in the development of downstream applications**.
> - **First attempt to instruct-tune LLMs with agent trajectories across multiple tasks:** To the best of our knowledge, **we are the first to use agent trajectories from multiple tasks for instruction tuning**, achieving performance comparable to GPT-3.5-turbo. Furthermore, our error analysis (Cf Sec 3.3) shows that the lack of alignment is the main issue with the agent capabilities of models, which can be greatly improved with minimal data training. We believe this will provide insights into how to enhance the inherent agent capabilities of models.
> - **Automated and Scalable Trajectory Collection Methodology:** Given that agent trajectories are precious and challenging to collect, we have established an **automated and scalable method for collecting agent trajectories and have verified its effectiveness**. For the Instruction Generation process, we used two methods, Self-Instruct and Task Derivation, to supplement tasks lacking training instructions. This **ensures that we have sufficiently diverse agent trajectories**. In the Trajectory Collection process, we interact using GPT-4 and aggressively apply a filtering of $r=1$ to almost all trajectories, **ensuring we have high-quality agent trajectories**. Both of these steps **require no manual intervention and can be easily extended to other agent tasks**. By fine-tuning the model on 1,866 trajectories from 6 held-in tasks, there was a significant improvement in performance on 6 held-out (unseen) Agent tasks, demonstrating the effectiveness of our method.
>
> Regarding your concerns about the innovativeness of using the ReACT/CoT prompt method. We wish to emphasize that while previous works [4][5] have indeed innovated in prompting methods to enhance the performance of API-based models, **they have not focused on improving the agent capabilities of the models themselves**, which is the main focus of our paper. Considering the fact that open-source models are far behind API models in agent capabilities, we have for the first time **elevated the general-purpose agent capability of an open-source model to the level of GPT-3.5-turbo**, even just by using the common ReACT prompt method. We consider this to be a significant contribution. Of course, we believe that more advanced prompting methods are very worthwhile to research, and we intend to make this our next direction of exploration.

---

> > ### Author Response · Authors · 2023-11-20
> > **Response to Reviewer iTiT (2/4)**
> >
> > > Weakness 2: Using an already available corpora with ReAct and filtering trajectories based on a threshold seems to be reducing the size of the data drastically. For example, in Mind2Web, only a tiny fraction is kept. It is not clear if the benefit of COT annotated trajectories can overcome the reduction in the data size. Can you present results where you make use of the data as much as possible even if you can’t inject COT annotations?
> >
> > We believe that **more diverse, higher-quality data** is more effective than a large amount of low-quality data, which is consistent with the conclusions in [7]. We think this is also a core factor why the AgentTuning method can **successfully generalize to unseen agent tasks while maintaining general capabilities**. We explain our approach as follows:
> >
> > - **Effectiveness of the Trejactory Filtering:** As demonstrated in Table 2 (Cf Sec. 2.1.3), the abliation results with filtered trejactories are significantly better than using as much data as possible (unfiltered trejactories). This verifies that using a small amount of high-quality data is actually better than training with all available data.
> > - **Effectiveness of the CoT rationale:** We'd like to clarify that all trajectories interacted by GPT-4 have CoT rationale except constructed using Task Derivation. For trajectories constructed via Task Derivation, we also use GPT-4 to supplement the reasoning process. We have also conducted a set of comparative experiments showing that training with the CoT process yields better performance for the model.
> >
> > |  | AgentLlama-7B (w/o Filter) | AgentLlama-7B (w/o CoT) | AgentLlama-7B |
> > | :---: | :---: | :---: | :---: |
> > | Held-in | 1.34 | 1.38 | **1.96** |
> > | Held-out | 0.47 | 0.56 | **0.65** |
> >
> > > Weakness 3: How did you choose η? Given that it denotes the tradeoff between broader generalization vs agent-task performance, it is important to highlight its impact. Are your models sensitive to this parameter?
> >
> > The hyper-parameter $\eta$ signifies the balance between the general language abiliy and agent task ability. We conducted grid searches for $\eta$ on 7B and 13B scales. We use the scores on held-in agent task, general tasks and their average as evalutaion metrics. Thus, no information about held-out task is leaking through this hyper-parameter. The scores of held-in tasks and general tasks are not directly comparable. To compose the average score, all scores are normalized with the maximum score in each category (i.e. the score of GPT-4).
> >
> > - Grid Search in 7B scale
> >
> >     |$\eta$|0.0|0.1|0.2|0.3|0.4|0.5|0.6|0.7|0.8|0.9|1.0|
> >     |---|---|---|---|---|---|---|---|---|---|---|---|
> >     |7B Held-in| 0.13 | 0.72 | 0.68 | 0.72 | 0.73 | 0.74 | 0.72 | 0.75 | 0.74 | 0.69 | 0.50 |
> >     |7B General| 0.42 | 0.42 | 0.41 | 0.41 | 0.40 | 0.40 | 0.40 | 0.41 | 0.37 | 0.36 | 0.14 |
> >     |7B Average| 0.27 | **0.57** | 0.55 | **0.57** | 0.56 | **0.57** | 0.56 | **0.58** | 0.55 | 0.53 | 0.32 |
> >
> > - Gird Search in 13B scale
> >
> >     |$\eta$|0.0|0.1|0.2|0.3|0.4|0.5|0.6|0.7|0.8|0.9|1.0|
> >     |---|---|---|---|---|---|---|---|---|---|---|---|
> >     |13B Held-in| 0.15 | 0.72 | 0.77 | 0.74 | 0.61 | 0.66 | 0.70 | 0.60 | 0.67 | 0.72 | 0.57 |
> >     |13B General| 0.45 | 0.45 | 0.47 | 0.45 | 0.47 | 0.47 | 0.46 | 0.47 | 0.42 | 0.35 | 0.12 |
> >     |13B Average| 0.30 | **0.59** | **0.62** | **0.59** | 0.54 | 0.56 | 0.58 | 0.54 | 0.54 | 0.54 | 0.35 |
> >
> > The table presents how different $\eta$ values (ranging from 0 to 1 with increments of 0.1) affect the performance of models on 7B and 13B scales. As demonstrated in the table, the model is not very sensitive to $\eta$, except for the extremes values of $\eta$=0 and $\eta$=1, the performance of which has already been reported in Table 5 of the paper. The average score seems evenly distributed in 7B models and forms a peak at 0.2 in 13B models. Thue we select the best $\eta$ in 13B models, i.e. 0.2, as the final $\eta$ for all models. (Clearify an error in Appendix A. We stated that $\eta$=0.4 is used for AgentLlama-7B. This is a mistake and will be corrected in a later version. All results are reported with AgentLlama models trained from $\eta$=0.2 .) The above results will be included in the appendix for a better understanding of hyper-parameter selection.

---

> > > ### Author Response · Authors · 2023-11-20
> > > **Response to Reviewer iTiT (3/4)**
> > >
> > > > Weakness 4: Similarly, how did you decide 1:4 ratio for GPT-4 vs GPT-3.5?
> > >
> > > The sampling ratio between GPT-4 and GPT-3.5 is selected by grid searching on 7B scale, varing from 0.0 to 0.5 with incrementent 0.1. Only the score of general tasks is meausred, as this sampling ratio signifies the trade-off between the quality and diversity of the general data.
> > >
> > > | GPT-4 Sampling Ratio| 0.5  | 0.4  | 0.3  | 0.2  | 0.1  | 0.0  |
> > > | --------- | ---- | ---- | ---- | ---- | ---- | ---- |
> > > | MMLU      | 48.4 | 48.4 | 49.3 | 48.7 | 46.0 | 49.3 |
> > > | HumanEval | 14.4 | 14.9 | 11.3 | 15.4 | 6.20 | 5.50 |
> > > | MT-Bench  | 6.00 | 6.08 | 6.15 | 6.34 | 6.40 | 6.18 |
> > > | GSM8K     | 21.8 | 24.6 | 23.7 | 24.6 | 23.7 | 26.5 |
> > > | **AVG**   | 0.60 | 0.62 | 0.59 | **0.63** | 0.55 | 0.57 |
> > >
> > > As indicated by the grid search, the model is not very sensitive to this sampling ratio and we choose the value where we have the maximum averge score, i.e. 0.2 (GPT4 : GPT3.5 = 1 : 4). We will the add the above experimental results into the Appendix of our paper for a comprehensive understanding of how the sampling ratio is selected.
> > >
> > > > Weakness 5: What is GPT-4’s interaction trajectory with the DB? How did you end up collecting multi-turn dialogues? Are you using rule-based turn generation to convert DB response into a prompt?
> > >
> > > 1. **Definition of the term "interaction trajectory":** We use the term "interaction trajectory" to refer the total conversation history between the agent and the environment.
> > >
> > >     In AgentBench settings, the framework will serve as an intermediate layer between the agent and the database. In the first turn, the framework prepares database environment specified by a task example and prompts the agent with the task description, and then each turn the agent response with a thought and a SQL statement to execute, or a final answer if the agent determines that the task has been solved. The framework execute the SQL statement in the database and prompts the agent with the result returned by the database, along with the conversation history.
> > >
> > >     So, by the term "interaction trajectory with the database", we mean **all conversations prompted to and responded by the agent**, which contains not only the statements executed in the database and the corresponding results, but also the task description, the agent's thoughts, etc.
> > >
> > > 2. **Collection of multi-turn dialogues:** We collected multi-turn dialogues from the interaction trajectories of GPT-4, since some tasks are solved in multiple steps.
> > >
> > >     As explained in *Section 2.1.1 Instruction Construction* of our paper, there are 3 kinds of different DB trajectories used in our training:
> > >
> > >     1. Task derivation trajectories synthesized from BIRD. Using questions and corresponding reference SQL answers in BIRD dataset, we manually construct a 2-round conversation with the agent executing the SQL statement in the first round, and submitting the answer in the second round. We then show the conversation to GPT-4 and query GPT-4 to fill in the thoughts in a Chain-of-Thought manner in order to better enhance AgentLlama's reasoning ability.
> > >     2. Task derivation trajectories by GPT-4 collected from BIRD. Noticing that the above manner can only generate conversation trajectories of fixed rounds, we then prompt GPT-4 with questions from BIRD and collect its multi-round conversation trajectories to enhance AgentLlama's multi-turn reasoning ability.
> > >     3. Self-Instruct trajectores by GPT-4. As BIRD only contains SQL SELECT tasks, we utilize the Self-Instruct method to generate other types of SQL tasks, namely INSERT, UPDATE and DELETE. We first ask GPT-4 to generate these types of tasks, then we ask GPT-4 to solve this task in a newly initiated conversation and collected the trajectories.
> > >
> > >     Among the above 3 types of DB trajectories, type 1 is synthesized and is fixed at 2 turns, and type 2 and 3 are GPT-4's interaction trajectories so **they are multi-turn since some tasks require multiple steps to solve**.
> > >
> > > 3. **Use of rule-based generation:** Type 1 is partially rule-based generated since the conversations are synthesized directly from the task and the answer, with the thoughts filled by GPT-4. Type 2 and 3 are not rule-based generated. Instead, they are collected from GPT-4 trajectories automatically.
> > >
> > >     As for the DB response, according to the setting of AgentBench, it is always provided to the agent verbatim. That is, we are not converting the execution result of the database into other forms, e.g. into natural language. We just feed the result back to the agent as its observation from the environment.

---

> > > > ### Author Response · Authors · 2023-11-20
> > > > **Response to Reviewer iTiT (4/4)**
> > > >
> > > > > Weakness 6: Is ShareGPT not leaking any of the test-time task data? It would be helpful to clarify.
> > > >
> > > > Thanks for your feedback regarding the data leakage concern, which underscores the importance of data integrity in our research. We are grateful for your dedication to upholding high standards.
> > > >
> > > > After additional data contamination analysis, the answer is that **ShareGPT is not leaking the test-time task data**.
> > > >
> > > > We've already conducted data contamination analysis of our AgentInstruct dataset in *Appendix B Data Contamination* of our paper, with a token-based approach following [6]. We didn't analyze the ShareGPT dataset in the first version of our paper because general domain datasets like ShareGPT have demonstrated a lack of agent task data, which results in models trained solely on these datasets performing poorly in agent tasks. This is also our main motivation to introduce the AgentInstruct dataset which consists of high-quality agent task data.
> > > >
> > > > To verify that our use of ShareGPT data does not incur data leakage, we conducted the same data contamination analysis in terms of ShareGPT data used in our training, and found **no indications of data leakage**. The results are presented in the table below.
> > > >
> > > > **Data Contamination Analysis of ShareGPT Data**
> > > >
> > > > | Task | Contamination Rate (%) | # Clean Ex. | # Dirty Ex. | # Examples |
> > > > | :---: | :---: | :---: | :---: | :---: |
> > > > | ALFWorld | 0.00  | 40 | 0 | 50 |
> > > > | Database | 0.00  | 300 | 0 | 300 |
> > > > | KnowledgeGraph | 2.14  | 144 | 0 | 150 |
> > > > | Mind2Web | 10.72  | 153 | 0 | 177 |
> > > > | OperatingSystem | 0.00  | 88 | 0 | 144 |
> > > > | WebShop | 22.87  | 77 | 0 | 200 |
> > > > | Total | 5.41  | 802 | 0 | 1021 |
> > > >
> > > > Here, we define a test example as "clean" if it has under 20% tokens contaminated (included in a 10-gram that appears in both the test example and the training set), and "dirty" if it has over 80% tokens contaminated. These definitions are taken from [6]. Please refer to Appendix B for more details.
> > > >
> > > > Most tasks have a nearly zero contamination rate and have most test examples clean. Mind2Web and WebShop have relatively higher contamination rates due to their task description style, as we explained in Appendix B. However, there are no dirty examples in these test sets, which means that the test examples themselves as a whole do not appear in the training set (which we call a data leakage), just some phrases in task descriptions do.
> > > >
> > > > Overall, no dirty examples are observed in the test set, which is a strong signal that **the ShareGPT data is not leaking any of the test-time task data**.
> > > >
> > > >
> > > > ### References
> > > >
> > > >
> > > > [1] Deng, Xiang, et al. "Mind2Web: Towards a Generalist Agent for the Web." arXiv preprint arXiv:2306.06070 (2023).
> > > >
> > > > [2] Yao, Shunyu, et al. "Webshop: Towards scalable real-world web interaction with grounded language agents." Advances in Neural Information Processing Systems 35 (2022): 20744-20757.
> > > >
> > > > [3] Qin, Yujia, et al. "Toolllm: Facilitating large language models to master 16000+ real-world apis." arXiv preprint arXiv:2307.16789 (2023).
> > > >
> > > > [4] Xu, Binfeng, et al. "ReWOO: Decoupling Reasoning from Observations for Efficient Augmented Language Models." arXiv preprint arXiv:2305.18323 (2023).
> > > >
> > > > [5] Shinn, Noah, et al. "Reflexion: Language Agents with Verbal Reinforcement Learning, June 2023." arXiv preprint arXiv:2303.11366.
> > > >
> > > > [6] Touvron, Hugo, et al. "Llama 2: Open foundation and fine-tuned chat models." arXiv preprint arXiv:2307.09288 (2023).
> > > >
> > > > [7] Zhou, Chunting, et al. "Lima: Less is more for alignment." arXiv preprint arXiv:2305.11206 (2023).

---

> ### Comment · Reviewer_iTiT · 2023-11-21
> **Thank you for the update.**
>
> Thank you for the detailed response and additional experiments. It helped clarify many of my concerns. I raised my score accordingly.
>
> I would like to further discuss my point about weakness 2. I am curious about just fine-tuning Llama on all "originally available training data that a task has". For example, WebAgent [1] uses all available trajectories from WebShop to fine-tune a T5 model. It is not a generalist agent and it doesn't utilize CoT rationales but it can use all available training data. I think the problem stems from using GPT to collect interaction trajectories; since GPT can lead to unsuccessful trajectories, without filtering them you would be training on unsuccessful trajectories; which could also be why you get worse result without filtering. An alternative would be using GPT to **supplement every trajectory that a task originally has with thoughts using ReAct prompting, instead of collecting trajectories using GPT**. My understanding is that you do this only for trajectories generated using task derivation but please clarify if I am mistaken.
>
> One additional thing about **training corpus for AgentLlama-7B (w/o CoT)**. Did you use the same set of demonstrations as in AgentLlama-7B but just removed CoT rationales?
>
> [1] A Real-World WebAgent with Planning, Long Context Understanding, and Program Synthesis, Gur et al. https://arxiv.org/abs/2307.12856

---

> > ### Author Response · Authors · 2023-11-23
> > **Additional Response to Reviewer iTiT (1/3)**
> >
> > Thank you once again for your insightful feedback and constructive suggestions. We deeply appreciate your continued engagement and the valuable perspectives you bring to our work.
> >
> > Concerning annotating thoughts on the training set, we added a supplementary experiment within the rebuttal time budget, and **it turns out that direct thought annotation on demonstrations yields sub-optimal results owing to perception discrepancies between human annotators and LLMs**. We also detailed how we generated the AgentInstruct(w/o CoT) corpus.
> >
> >
> > ### Annotating Thoughts on Training Set
> >
> > > I am curious about just fine-tuning Llama on all "originally available training data that a task has". For example, WebAgent [1] uses all available trajectories from WebShop to fine-tune a T5 model. It is not a generalist agent and it doesn't utilize CoT rationales but it can use all available training data. I think the problem stems from using GPT to collect interaction trajectories; since GPT can lead to unsuccessful trajectories, without filtering them you would be training on unsuccessful trajectories; which could also be why you get worse results without filtering. ...
> >
> > Annotating thoughts on the training set / human demonstrations truly provides much more trajectories. We provide an extra experiment on your suggestion.
> >
> > **Experimental Setup**
> >
> > Not all tasks come with a readily available training set / human demonstration, we only used the human demonstration in WebShop and Mind2Web in this experiment.
> >
> > - **Preprocess.** We preprocess the 1k human demonstration trajectories in WebShop and 1k human demonstration trajectories in Mind2Web to precisely follow the format we used in AgentInstruct.
> > - **Reward Filtering.** We notice that some human trajectories in the WebShop training set have reward < 1, we only kept 775 trajectories where reward = 1.
> > - **Data Re-balance.** Mind2Web and WebShop have significantly more samples, direct concatenation leads to Mind2Web and WebShop being learned too much while other tasks are starving. We decided to use a sampling ratio strategy to compensate for this imbalance. We first calculate the original sampling rate in *AgentInstruct* and we slightly improve the sampling ratio to be over 10% in Mind2Web since we have much more data now. The sampling ratio of WebShop is left unchanged since it is already relatively large among the tasks. The relative ratios of other tasks are also unchanged to make the results comparable with AgentLlama-7B. The sampling ratios are then normalized 100%, forming *AgentInstruct(more)*.
> >
> >     |agent dataset|DB|WebShop|ALFWorld|KG|OS|Mind2Web|
> >     |---|---|---|---|---|---|---|
> >     |AgentInstruct|28.8%|18.8%|18.0%|17.4%|10.5%|6.5%|
> >     |AgentInstruct (more)|27.7%|18.1%|17.3%|16.7%|10.1%|10.1%|
> >
> >     The sampling ratio between *AgentInstruct(more)* as a whole and all general data is also set the same as AgentLlama-7B.
> >
> > **Preliminary Results**
> >
> >
> >
> > | Held-in tasks             | ALFWorld | WebShop | Mind2Web | Knowledge Graph | Operating System | Database | AVG/Ours |
> > | ------------------- | -------- | ------- | -------- | --------------- | ---------------- | -------- | -------- |
> > | Llama2-7B                  | 2.0      | 4.4     | 3.7      | 0.0             | 8.3              | 0.3      | 0.16     |
> > | AgentLlama-7B          | **84.0** | **63.6**    | 6.4      | 18.1            | **17.4**         | 30.6     | **1.61** |
> > | AgentLlama-7B (more) | 80.0     | 11.2    | **9.9**  | **25.7**        | 14.6             | **31.0** | 1.43     |
> >
> > | Held-out tasks           | SciWorld | MiniWoB++ | WebArena | HotpotQA | ReWOO    | Digital Card Game | AVG      |
> > | ------------------- | -------- | --------- | -------- | -------- | -------- | ----------------- | -------- |
> > | Llama2-7B                  | 5.9      | 0.0       | 1.2      | 22.6     | 48.3     | 0.0               | 0.35     |
> > | AgentLlama-7B          | 13.7     | **28.9**  | 0.7      | **22.3** | **50.9** | **7.0**           | **0.58** |
> > | AgentLlama-7B (more) | **14.5** | 19.6      | 0.7      | 21.2     | 33.1     | 6.0               | 0.49     |
> >
> > | General tasks            | MMLU     | HumanEval | MT-Bench | GSM8k    | AVG   |
> > | ------------------- | -------- | --------- | -------- | -------- | ----- |
> > | Llama2-7B                  | 48.0     | 13.9      | 6.26     | **27.7** | 0.583 |
> > | AgentLlama-7B          | **48.7** | **15.4**  | 6.34     | 24.6     | **0.585** |
> > | AgentLlama-7B (more) | 48.6     | 13.0      | **6.50** | 26.2     | 0.582 |
> >
> > Compared with AgentLlama-7B, AgentLlama-7B (more) is slightly improved in Mind2Web but significantly worse in WebShop. Its performance in general tasks can be viewed as unchanged, and its score slightly drops in held-out tasks.

---

> > > ### Author Response · Authors · 2023-11-23
> > > **Additional Response to Reviewer iTiT (2/3)**
> > >
> > > **Analysis**
> > >
> > > 1. **Worse performance in WebShop.** After an examination of the annotations, we discover that GPT-4 does not understand many of the human annotations trajectories, even with reward = 1. The following is an example that confuses GPT-4.
> > >
> > >
> > >     | Action                  | Description                                                  | GPT-4 Thought                                                |
> > >     | ------------------------ | ------------------------------------------------------------ | ------------------------------------------------------------ |
> > >     | (Instruction)              | i would like to get some size 6.5 **yellow non slip flip** flops, and price lower than 130.00 dollars | /                                                            |
> > >     | search                   | **women's pumps**                                                | I'm going to search for it online to see what options are available. |
> > >     | click                    | B09SNV5NK5                                                   | I decided to click on 'Woman Slippers' to find out more details, including whether it is available in yellow, non-slip, and in size 6.5 as per my requirements. |                                                           |
> > >     | click | Buy now                                                      | **I didn't get enough information about the product and it doesn't seem to fit my criteria.** I don't see any information about the size, color, or whether it's non-slip. Next, I'll click on the 'Buy Now' button since that seems to be the correct action even though it's not clear why I'm buying the wrong item.|
> > >     | (Reward provided in the dataset)     | 1.0                                                          | /                                                            |
> > >
> > >     After a deeper investigation, we notice that the "confusion" might stem from the modality discrepancy between human annotators and LLMs. In the above example, the human annotator chooses *women's pumps* mainly for its **yellow color** and it **looks just like a flip flop** (Image for "*women's pumps*" https://i.imgur.com/8AjaM91.jpeg, which is seen by human-annotators but not LLMs). In contrast, the LLM can only interpret linguistic contexts, making it unable to annotate some results. Hence, thoughts annotated on human trajectories fail to align directly with what the LLM can see, which leads to many misleading annotations.
> > >
> > > 2. **Slight improvement in Mind2Web.** Although with much more demonstration, the improvement in Mind2Web is modest. Inspection of CoT annotations also shows that the thought-annotation model fails to understand the rationales behind many of the human demonstrations. Except for the lack of visual information, LLMs can only perceive a highly simplified DOM tree in the setting of this benchmark, but a complete DOM tree is rendered for the human annotators. This makes the thought-annotation model unable to understand some of the human actions.
> > >
> > > 3. **Worse performance in held-out.** Analysis on held-out reveals that the model's ability to follow instructions decreases and performs more trivial errors in choosing the action to take, which leads to more failed results. We believe that the "confusing" GPT-4 annotations cripple the CoT reasoning ability of the model, making it unable to reason the correct action to take.
> > >
> > >
> > > In short, directly injecting annotations into complex task trajectories like WebShop and Mind2Web does not bring about significant performance improvement, probably because the thought-annotation model fails to obtain the visual information and the information excluded by the benchmarking environment, which is available to human annotators. We will add a detailed analysis of the information gap between human interaction trajectories and LLM trajectories in the paper. We will also add experiments across scales with an improved thought annotation pipeline to exclude the sample affected by the perception gap between LLM and humans (in the camera-ready version if accepted). Thanks again for this constructive suggestion on our method.
> > >
> > > ---

---

> > > > ### Author Response · Authors · 2023-11-23
> > > > **Additional Response to Reviewer iTiT (3/3)**
> > > >
> > > > ### Generation Details of w/o CoT Data
> > > >
> > > > > One additional thing about training corpus for AgentLlama-7B (w/o CoT). Did you use the same set of demonstrations as in AgentLlama-7B but just removed CoT rationales?
> > > >
> > > > We use the same 1,866 trajectories that are used in the AgentInstruct dataset. Further, we notice that directly removing the CoT rationales makes the context inconsistent for language models to follow. For the training corpus for AgentLlama-7B (w/o CoT), we rewrite all instructions to remove the instructions for generating thoughts, and we remove the thoughts in both few-shot prompting and demonstrations accordingly.
> > > >
> > > > For instance, in the OS task, modifications in the instruction and response are marked as strikethrough.
> > > >
> > > > **Instruction**
> > > >
> > > > > You are an assistant that will act like a person, I'will play the role of linux(ubuntu) operating system. Your goal is to implement the operations required by me or answer to the question proposed by me. For each of your turn, you should first think what you should do, and then take exact one of the three actions: "bash", "finish" or "answer".
> > > > >
> > > > > 1. If you think you should execute some bash code, take bash action, and you should print like this:
> > > > >
> > > > > **~~Think: put your thought here.~~**
> > > > >
> > > > > Act: bash
> > > > >
> > > > > ```bash
> > > > > # put your bash code here
> > > > > ```
> > > > >
> > > > > 2. If you think you have finished the task, take finish action, and you should print like this:
> > > > >
> > > > > **~~Think: put your thought here.~~**
> > > > >
> > > > > Act: finish
> > > > >
> > > > > 3. If you think you have got the answer to the question, take answer action, and you should print like this:
> > > > >
> > > > > **~~Think: put your thought here.~~**
> > > > >
> > > > > Act: answer(Your answer to the question should be put in this pair of parentheses)
> > > > >
> > > > > If the output is too long, I will truncate it. The truncated output is not complete. You have to deal with the truncating problem by yourself. Attention, your bash code should not contain any input operation. Once again, you should take only exact one of the three actions in each turn.
> > > > >
> > > > > Now, my problem is:
> > > > >
> > > > > tell me how many files are in the directory "/etc"?bash:`
> > > >
> > > > **Response**
> > > >
> > > > > **~~Think: To count the files in /etc, I need to print all the files in it.~~**
> > > > >
> > > > > Act: bash
> > > > >
> > > > > ```bash
> > > > > ls /etc
> > > > > ```
> > > >
> > > > Other tasks in AgentInstruct(w/o CoT) are processed similarly.
> > > >
> > > > ---
> > > >
> > > > Your feedback is invaluable to us, and we are committed to continual improvement and excellence in our research endeavors.  If this response has addressed your concerns, may we respectfully inquire about the potential for an improved score on our submission?

---

### Official Review · Reviewer_cRHd · 2023-11-02

**Soundness:** 2 fair
**Presentation:** 3 good
**Contribution:** 4 excellent
**Rating:** 6
**Confidence:** 4

**Summary:**

The authors present to fine-tuning LLMs for agent behaviors with a new dataset collected using demonstrations of GPT models, and demonstrated that LLAMA models achieve significantly better performance on the held-out test set after fine-tuning.

**Strengths:**

1. Agent tuning is an exciting and important direction to study for the LLMs as intelligent agents.
2. The authors' data/training/model have been well-documented. The results should be reproducible

**Weaknesses:**

1. Figure 1 (b), I don't think the message is fair for this figure, since you trained on AgentBench (although partly), but the other LLMs have not trained on AgentBench. One of the down-sides for open-source LLMs is the ability to generalize to `different' settings from training, but the proposed work has essentially made AgentBench in-distribution by training.
2. It seems that GPT models are heavily relied on for generating training data. Do we have some sense of how to go beyond GPT models? Suppose we want to push the boundaries of GPT-4, then GPT-4 data may not work as well.
3. I have concerns for generalization to other agent tasks that are distinct, but not captured in agent-bench. For example, driving or operating Minecraft.

Overall, my main concern is that the improvements might have come from better instruction following, not agent reasoning.

Minor issues:
- Figure 1 (a), Where does the overall score stand against GPT-4, from which you collected training data?

**Questions:**

1. How does AlfWorld results in table 4 compare to [1], which reported higher score than the highlighted best open-source model?




[1] Micheli, Vincent, and François Fleuret. "Language models are few-shot butlers." arXiv preprint arXiv:2104.07972 (2021).

---

> ### Author Response · Authors · 2023-11-20
> **Response to Reviewer cRHd  (1/2)**
>
> We would like to express our sincere gratitude for your insightful comments and constructive criticism regarding our paper. Your feedback is invaluable in enhancing the quality and clarity of our research. We understand your concerns and are happy to provide further explanations as follows:
>
> > Weakness 1: Figure 1 (b), I don't think the message is fair for this figure, since you trained on AgentBench (although partly), but the other LLMs have not trained on AgentBench. One of the down-sides for open-source LLMs is the ability to generalize to \`different\' settings from training, but the proposed work has essentially made AgentBench in-distribution by training.
>
> Firstly, it is important to emphasize that AgentBench was not the primary benchmark used to validate the agent capabilities of our model. The purpose of Figure 1 (b) was primarily to **illustrate the performance gap between open-source models and API-based models**.
>
> Moreover, we have conducted **systematic evaluation on 6 held-out agent tasks**, which was not included in the training set. This demonstrate the robust generalization performance of our models. The results from these tests, as detailed in our paper (Cf Sec. 3.2), show that our model exhibits strong agent capabilities on held-out agent tasks.
>
> > Weakness 2: It seems that GPT models are heavily relied on for generating training data. Do we have some sense of how to go beyond GPT models? Suppose we want to push the boundaries of GPT-4, then GPT-4 data may not work as well.
>
> The use of GPT-4 generated data in our experiments was a strategic choice, primarily for its convenience of data generation and the quality of the trajectories.
>
> We believe that AgentTuning can serve as a **foundational method that can be readily combined with other data-centric approaches to foster improvements that are not dependent on the external models like GPT-4**. For instance, a key step in AgentTuning is trajectory filtering, which rejects trajectories with poor performace with environment reward. Self-instruct may be combined with trajectory filtering to perform AgentTuning, where a model can be iteratively self-improving, like [1], while preserving general abilities through hybird instruction tuning.
>
> > Weakness 3: I have concerns for generalization to other agent tasks that are distinct, but not captured in agent-bench. For example, driving or operating Minecraft.
>
> We deeply appreciate your feedback concerning our agent's generalization ability. Your insights are crucial in ensuring the robustness of our research. In fact, generalization ability is one of our main goals, and we carried out extensive experiments to show that **our models hold strong generalization ability**.
>
> As described in our evalution setup (Cf Sec. 3.1), we conducted experiments on 6 held-out tasks that are not covered in our training set, with **5 of them not from AgentBench**. The tasks vary greatly in their respective domains. Specifically, the 6 tasks cover tasks of digital card games, daily computer tasks, science experiments, web interaction, wiki retrieval and question answering. As shown in *Section 3.2 Main Results*, our models **achieve huge improvement on these held-out tasks despite they have never seen them before**, with our 70B model getting +176% improvement in the overall score of held-out tasks, which nearly matches the score of GPT-3.5. This clearly shows that our model can generalize to other unseen agent tasks well.
>
> Playing Minecraft is a rather hard task that **even GPT-3.5 performs poorly** according to [3]. They carried out experiments and found out that GPT-3.5 and open-source LLMs cannot deal with playing Minecraft well. In their experiments, substituting GPT-4 with GPT-3.5 in code generation leads to a sharp performance degradation, with over 80% drop in number of unique items obtained.
>
> Besides, [3] utilizes a set of complicated prompts to enable GPT-4 to play the game effectively. However, our work focuses on **improving the inherent agent ability of models**, which will naturally benefit from improvements on prompting methods.
>
> Therefore, in current stage, we did not test our models on these highly difficult tasks. Instead, we leave further enhancing the agent ability of LLMs and finally enabling LLMs to tackle these hard tasks as a future working direction.

---

> ### Author Response · Authors · 2023-11-20
> **Response to Reviewer cRHd (2/2)**
>
> > Overall, my main concern is that the improvements might have come from better instruction following, not agent reasoning.
>
> In our work on AgentTuning, our primary goal was to enhance the performance of LLMs in generalized, unseen agent tasks. Through our experiments, we observed a notable improvement in the instruction-following capabilities, as demonstrated in Figure 3(a). This enhancement in instruction following for the Llama-2-chat models, was significant in reducing simple errors that are often lacking in many open-source models. We believe that this improvement in instruction following, which is essential for the successful completion of agent tasks, stands as a significant contribution of our work.
>
> Regarding agent reasoning, we acknowledge its importance in the interaction of LLMs with real world scenarios. However, we think agent reasoning is hard to precisely evaluated. As observed the held-out tasks, AgentTuning has enhanced the generalized agent performance of LLMs on unseen tasks. We welcome this opportunity for an open discussion to further explore and clarify this concept.
>
> > Figure 1 (a), Where does the overall score stand against GPT-4, from which you collected training data?
>
> Our main results in paper show both GPT-4's overall score in both held-in/held-out tasks (Cf Tab. 4). As we choose Llama 2 as our base model, which is a strong open-source model with capabilities comparable to GPT-3.5-turbo in general domains, but there is a huge gap between it and GPT-3.5-turbo in agent tasks. We believe it's possible to surpass GPT-4 using open-sourced LLMs in a specific agent task. Howerver, **as our focus is on general agent capabilities**, we think that using GPT-3.5-turbo as a baseline for comparison of agent capabilities is reasonable.
>
> > Question 1: How does AlfWorld results in table 4 compare to [1], which reported higher score than the highlighted best open-source model?
>
> The ALFWorld evalutaion setting in [1] is different from the evalution setting used in AgentTuning (which follows the setting in an [2]), so the evaluation scores are not directly comparable. Also, the model in [1] used task specific fine-tuning from a pre-trained model and can only solve ALFWorld task. That model probably has **no capability to follow general instructions and cannot perform generalized agent tasks**. This also makes this model unsuitale for a direct comparason.
>
>
> ### References
>
> [1] Micheli, Vincent, and François Fleuret. "Language models are few-shot butlers." arXiv preprint arXiv:2104.07972 (2021).
>
> [2] Liu, Xiao, et al. "Agentbench: Evaluating llms as agents." arXiv preprint arXiv:2308.03688 (2023).
>
> [3] Wang, Guanzhi, et al. "Voyager: An open-ended embodied agent with large language models." arXiv preprint arXiv:2305.16291 (2023).

---

> > ### Comment · Reviewer_7D3A · 2023-11-23
> >
> > I appreciate the authors for the additional experiments and further clarification. I think my concerns are mainly addressed and I choose to keep the points.

---

### Official Review · Reviewer_7D3A · 2023-11-03

**Soundness:** 3 good
**Presentation:** 3 good
**Contribution:** 3 good
**Rating:** 6
**Confidence:** 4

**Summary:**

The paper presents AgentTuning, a methodology designed to enhance the capabilities of Large Language Models (LLMs) when they are employed as agents, while preserving their broader language understanding and generation abilities. It introduces AgentInstruct, a specialized instruction-tuning dataset, and an integrated instruction-tuning approach that combines it with publicly accessible instructions spanning various domains. Through experimental evaluations conducted on AgentLLaMA (instruction-tuned LLaMA2 series), the paper demonstrates that AgentTuning substantially improves the performance of LLMs in agent roles while maintaining the LLMs' foundational language understanding and generation capabilities. Notably, AgentLLaMA-70B exhibits comparable performance to GPT-3.5-turbo on unseen agent-related tasks.

**Strengths:**

- The motivation to improve the agent ability of open-sourced LLM is good.
- It is well-written and the idea it presents is clear.
- The evaluation is extensive and the results look promising.

**Weaknesses:**

- Some details of the dataset construction is unclear.
- The training strategy used for instruction-tuning is limited.
- The rationale behind some design choice needs more explanations.

**Questions:**

This paper proposes the instruction-tuning dataset and hyprid instruction-tuning to improve the agent ability of open-sourced LLM (i.e., LLaMA2). I think it is a significant contribution to the LLM community. However, I have some concerns as the following.
- The authors claim that they use a reward $r$ to filter out low-quality trajectories, but how this reward is calculated/generated is unclear to me. Besides, as the trajectories are generated by GPT3.5/4, I think the trajectories may contain some misleading information. I wonder whether there is a human evaluation of the correctness of the trajectories.
- It leverages a hybrid instruction-tuning strategy, which is a simple multi-task training in essence. Based on my understanding, improving the agent ability while preserving the general language understanding ability is close to a continual learning scenario. So I think it is important to discuss some typical training strategies like weight regularization[1] and parameter allocation[2] used in continual learning. And it would be better if you can further provide some results when such strategies are applied to see whether these strategies can benefit training.
- The rationale behind some design choices is not well-explained. For example, why is the ratio of sampling between GPT3.5 and GPT4 set as 1:4 and why is the threshold of reward $r$ is set as 2/3? Are there any rules or experimental explanations for these choices?

[1] Kirkpatrick et al. Overcoming catastrophic forgetting in neural networks. Proceedings of the National Academy of Sciences 2017.
[2] Serra et al.  Overcoming catastrophic forgetting with hard attention to the task. ICML 2018.

---

> ### Author Response · Authors · 2023-11-20
> **Response to Reviewer 7D3A (1/3)**
>
> Thank you for your valuable feedback and insightful questions regarding our manuscript. We appreciate the opportunity to clarify the methodology used in our study.
>
> > Question 1: The authors claim that they use a reward r to filter out low-quality trajectories, but how this reward is calculated/generated is unclear to me. Besides, as the trajectories are generated by GPT3.5/4, I think the trajectories may contain some misleading information. I wonder whether there is a human evaluation of the correctness of the trajectories.
>
> **Clarification on Reward Calculation** As we adopt 6 held-in agent tasks from AgentBench [1], the detailed reward calculation formula for each task could be found in appendix of AgentBench. Reward represents the completion status of a task, with values ranging between $[0, 1]$, where 1 indicates that the task is fully completed.
> For a quick clarification, we have summarized 6 held-in tasks as a table below:
>
> |   Task   |        Description        |                     Example                      |       Reward       |                      Reward Calculation                      |
> | :------: | :-----------------------: | :----------------------------------------------: | :----------------: | :----------------------------------------------------------: |
> | ALFWorld | Daily Household  Routines |                    Heat food                     |    Success Rate    |           If task is finished, r=1, otherwise r=0            |
> | WebShop  |      Online Shopping      |                   Buy a shirt                    |       Reward       |     Score for selecting the correct item during shopping     |
> | Mind2Web |    Website Navigation     |                  Book a ticket                   | Step Success  Rate | Evaluate the predicted action correctness compared to reference actions. |
> |    KG    | Retrieve Entity from  KG  | Which team won the 2014 AFC  Championship Game?  |         F1         | Compare the model’s predicted answers to the gold standard answers |
> |    DB    |   Database  Operations    | How many games did the badgers  play in october? |    Step Success    |           If MySQL query is correct, r=1, otherwise r=0            |
> |    OS    |    Interacting with OS    |               Count specific files               |    Step Success    |           If result from operating system is correct, r=1, otherwise r=0            |
>
> **Human Evaluation and Dataset Verification** We fully understand your concern about the correctness of the trajectories, and we will explain the effectiveness of our data construction method without the need for manual intervention, as well as how we ensure the quality of the data. Since we have used trajectory filtering with $r=1$ for most tasks (except for MindWeb due to its difficulty), we can **ensure the correctness of the trajectories in these tasks**. Combining the Self-Instruct and Task Derivation methods for Instruction Generation, our data construction process **does not require manual participation, making our method scalable and easily extendable to other agent tasks**. Table 2 in the paper demonstrates that using trajectory filtering to eliminate incorrect trajectories significantly enhances the model's capabilities in Agent tasks. As for using GPT-4 to generate the Chain of Thought (CoT) reasoning process, its effectiveness has been verified by many works [2][3], and we have also conducted a set of comparative experiments showing that training with the CoT process yields better performance for the model.
>
> |  | AgentLlama-7B (w/ CoT) | AgentLlama-7B (w/o CoT) |
> | :---: | :---: | :---: |
> | Held-in | **1.96** | 1.38 |
> | Held-out | **0.67** | 0.56 |
> | General | **0.63** | 0.57 |

---

> ### Author Response · Authors · 2023-11-20
> **Response to Reviewer 7D3A (2/3)**
>
> > Question 2: It leverages a hybrid instruction-tuning strategy, which is a simple multi-task training in essence. Based on my understanding, improving the agent ability while preserving the general language understanding ability is close to a continual learning scenario. So I think it is important to discuss some typical training strategies like weight regularization[1] and parameter allocation[2] used in continual learning. And it would be better if you can further provide some results when such strategies are applied to see whether these strategies can benefit training.
>
> Thank you very much for your valuable suggestion regarding the discussion of typical continual learning strategies with AgentTuning methodology. We certainly agree that discussing and comparing our approach with established continual learning methods like weight regularization and parameter allocation would enrich the paper.  However, **there are specific challenges in directly applying many continual learning techniques to the fine-tuning of large language models (LLMs)**.
>
> For example, Elastic Weight Consolidation (EWC) necessitates access to the original supervised fine-tuning (SFT) data from the first task to estimate Fisher information for each parameter. Similarly, Hard Attention to the Task (HAT) requires the original SFT data to learn a task embedding with the original SFT data and modifications to the model architecture, such as adding gates between layers. However, the fine-tuning data for many open-source LLMs, like the Llama series, is not typically available. (Still, it's not totally impossible to combine these methods. For EWC, the fisher information of each parameter could be estimated with extra general task data like shareGPT. For HAT, the task embedding may also be esitmated through extra data also.)
>
> Another challenge the of implementing EWC/HAT entails the techniques used to train language models with huge parameters, like model/pipeline parallelism, distributed optimizers, etc. It requires huge modifications to Megatron training infrastructures (for example, reducing over the square of gradients, and accessing all parameters for EWC loss terms calculation) and would likely necessitate considerable effort in tuning method-specific hyper-parameters.
>
> Understanding the value of your suggestion to include discussions and comparisons with continual learning methods, we propose a different approach. Instead of integrating EWC and HAT methods, **we plan to explore LoRA and P-Tuning, both of which are prevalent and parameter-efficient fine-tuning methods in the realm of LLMs and are frequently applied in the continual learning of LLMs**. LoRA learns a low rank delta for the parameter matrices, while P-Tuning introduces an additional prefix in each layer. We will conduct experiments with these methods and include a comprehensive and detailed discussion in the Appendix. We are truly grateful for your constructive feedback and are confident that it will significantly enhance the depth and rigor of our study.

---

> > ### Author Response · Authors · 2023-11-20
> > **Response to Reviewer 7D3A (3/3)**
> >
> > > Question 3: The rationale behind some design choices is not well-explained. For example, why is the ratio of sampling between GPT3.5 and GPT4 set as 1:4 and why is the threshold of reward r is set as 2/3? Are there any rules or experimental explanations for these choices?
> >
> > Sorry for the confusion concerning the selection of sampling ratio and reward threshold.
> >
> > - **About sampling ratio between GPT-4 and GPT-3.5.** The data of GPT-4 and GPT-3.5 were added into training for preserving the general language ability of the model. The sampling ratio between GPT-4 and GPT-3.5 is selected by grid searching on 7B scale, varing from 0.0 to 0.5 with incrementent 0.1. Only the score of general tasks is meausred, as this sampling ratio signifies the trade-off between the quality and diversity of the general data.
> >
> >     | GPT-4 Sampling Ratio| 0.5  | 0.4  | 0.3  | 0.2  | 0.1  | 0.0  |
> >     | --------- | ---- | ---- | ---- | ---- | ---- | ---- |
> >     | MMLU      | 48.4 | 48.4 | 49.3 | 48.7 | 46.0 | 49.3 |
> >     | HumanEval | 14.4 | 14.9 | 11.3 | 15.4 | 6.20 | 5.50 |
> >     | MT-Bench  | 6.00 | 6.08 | 6.15 | 6.34 | 6.40 | 6.18 |
> >     | GSM8K     | 21.8 | 24.6 | 23.7 | 24.6 | 23.7 | 26.5 |
> >     | **AVG**   | 0.60 | 0.62 | 0.59 | **0.63** | 0.55 | 0.57 |
> >
> >     As indicated by the grid search, the model is not very sensitive to this sampling ratio and we choose the value where we have the maximum averge score, i.e. 0.2 (GPT4 : GPT3.5 = 1 : 4). We will the add the above experimental results into the Appendix of our paper for a comprehensive understanding of how the sampling ratio is selected.
> >
> > - **About the reward threshold 2/3** As mentioned in Section 2.1.3 of our paper, the reward filtering threshold for all held-in task are set as 1 (which means the result of the operation is totally correct), except Mind2Web, which is set as 2/3. Mind2Web task is particularly hard and even GPT-4 can only fully complete a small portion of tasks in the training set. This figure ([https://i.imgur.com/5G3MR2h.png](https://i.imgur.com/5G3MR2h.png)) shows the remaining number of trajectories under different filtering thresholds in the Mind2Web task. It can be observed that using a higher reward value for filtering results in a very small proportion of trajectories being retained. The filtering threshold for Mind2Web was chosen to strike a balance between data quality and diveristy whereas 2/3 is a relatively appropriate value.
> >
> > We are grateful for your constructive comments and hope that our response clarifies the methodology used in our study. Looking forward to any further suggestions or questions you might have.
> >
> > ### References
> >
> > [1] Liu, Xiao, et al. "Agentbench: Evaluating llms as agents." arXiv preprint arXiv:2308.03688 (2023).
> >
> > [2] Yue, Xiang, et al. "Mammoth: Building math generalist models through hybrid instruction tuning." arXiv preprint arXiv:2309.05653 (2023).
> >
> > [3] Mukherjee, Subhabrata, et al. "Orca: Progressive learning from complex explanation traces of gpt-4." arXiv preprint arXiv:2306.02707 (2023).

---

> > > ### Author Response · Authors · 2023-11-23
> > > **Thanks for your great review!**
> > >
> > > Dear Reviewer,
> > >
> > > Thank you very much again for your kind and valuable reviews! We hope our previous response addresses your questions, from which this work has benefited a lot. If possible, we would be deeply grateful if you could help share suggestions to further make AgentTuning a stronger work!
> > >
> > > Thank you again for your encouraging comments!
> > >
> > > Authors of Submission7956

---

### Meta-Review · Area_Chair_ystz · 2023-12-05

**Metareview:**

The paper presents "AgentTuning," a method to enhance the agent capabilities of LLMs by instruction-tuning on a dataset of high-quality interaction trajectories collected by GPT-3.5 and GPT-4 on AgentBench. The authors have open-sourced their dataset and fine-tuned Llama models, facilitating reproducibility and further research. Empirical results show comparable performance to GPT-3.5 on held-out tasks, showing this approach can generalize.

Weaknesses and Missing Elements:

- Dependence on GPT Models for Training Data: The heavy reliance on GPT models for generating training data raises questions about the method's applicability beyond these models and its capacity to push their boundaries.
- Incremental Nature of Improvements: The improvements, while empirically significant, are incremental advancements over existing instruction tuning and knowledge distillation methods.
- Limited tasks for training / evaluation: Compared to instruction tuning datasets such as FLAN that uses a large number of tasks, the paper focuses only on 6 agentic tasks for data collection during training. Evaluation is also limited to a few agent like tasks. For example, a reviewer evaluated the fine-tuned model on an [independent benchmark](https://github.com/microsoft/SmartPlay), and the improvements are not significant.

**Justification For Why Not Higher Score:**

All reviewers believe that this paper is marginally above acceptance threshold after discussions with the authors. However, the dependence of the paper's results on proprietary language models for training data collection and limited technical novelty are unaddressed.

Based on discussion with the SAC, I believe that paper's dataset contribution can be strengthened by removing dependence on GPT-* models.

**Justification For Why Not Lower Score:**

N/A

---

### Decision · Program_Chairs · 2024-01-16

Reject